# Cardiolipin for Enhanced Cellular Uptake and Cytotoxicity of Thermosensitive Liposome-Encapsulated Daunorubicin toward Breast Cancer Cell Lines

**DOI:** 10.3390/ijms231911763

**Published:** 2022-10-04

**Authors:** Hamad Alrbyawi, Sai H. S. Boddu, Ishwor Poudel, Manjusha Annaji, Nur Mita, Robert D. Arnold, Amit K. Tiwari, R. Jayachandra Babu

**Affiliations:** 1Department of Drug Discovery and Development, Harrison College of Pharmacy, Auburn University, Auburn, AL 36849, USA; 2Pharmaceutics and Pharmaceutical Technology Department, College of Pharmacy, Taibah University, Medina 42353, Saudi Arabia; 3Department of Pharmaceutical Sciences, College of Pharmacy and Health Sciences, Ajman University, Ajman P.O. Box 346, United Arab Emirates; 4Center of Medical and Bio-Allied Health Sciences Research, Ajman University, Ajman P.O. Box 346, United Arab Emirates; 5Department of Pharmacology and Experimental Therapeutics, The University of Toledo, Toledo, OH 43614, USA

**Keywords:** thermosensitive, liposomes, breast cancer, cardiolipin, daunorubicin

## Abstract

Daunorubicin (DNR) and cardiolipin (CL) were co-delivered using thermosensitive liposomes (TSLs). 1,2-dipalmitoyl-sn-glycero-3-phosphocholine (DPPC), 1-myristoyl-2-stearoyl-sn-glycero-3-phosphocholine (MSPC), cholesterol, 1,2-distearoyl-sn-glycero-3-phosphoethanolamine-N-[methoxy(polyethylene glycol)-2000] or DSPE-mPEG (2000) and CL were used in the formulation of liposomes at a molar ratio of 57:40:30:3:20, respectively. CL forms raft-like microdomains that may relocate and change lipid organization of the outer and inner mitochondrial membranes. Such transbilayer lipid movement eventually leads to membrane permeabilization. TSLs were prepared by thin-film hydration (drug:lipid ratio 1:5) where DNR was encapsulated within the aqueous core of the liposomes and CL acted as a component of the lipid bilayer. The liposomes exhibited high drug encapsulation efficiency (>90%), small size (~115 nm), narrow size distribution (polydispersity index ~0.12), and a rapid release profile under the influence of mild hyperthermia. The liposomes also exhibited ~4-fold higher cytotoxicity against MDA-MB-231 cells compared to DNR or liposomes similar to DaunoXome^®^ (*p* < 0.001). This study provides a basis for developing a co-delivery system of DNR and CL encapsulated in liposomes for treatment of breast cancer.

## 1. Introduction

Breast cancer is a leading cause of cancer mortality among women worldwide [1]. In the United States alone, more than 266,120 new breast cancer cases were diagnosed and 41,400 women died as the result of the disease in 2018 [2]. Based on the most recent data, the 5–year mortality rate for women diagnosed with breast cancer is 53% [3]. Surgery with or without radiotherapy achieves local control of cancer; however, when there is metastasis, systemic treatment is used in the form of hormonal therapy, chemotherapy, targeted therapy, or any combination of these [4,5]. Using chemotherapy for breast cancer treatment becomes ineffective in many patients and does not improve life expectancy as only a few patients with metastatic disease are cured, and treatments frequently cause significant adverse effects [6].

Chemotherapy is commonly administered if the tumor reaches a high grade and/or is node-positive [7]. Docetaxel, doxorubicin, cyclophosphamide, paclitaxel, and 5-fluorouracil are the most active cytotoxic agents for both early and advanced-stage breast cancer. Anthracycline or taxane drugs in combination with fluorouracil and cyclophosphamide are current therapeutics for breast cancer treatment [8]. Generally, cytotoxic drugs are highly toxic, nonspecific, and do not differentiate between healthy and cancerous cells. Chemotherapy-induced toxicities, such as neurotoxicity, cardiotoxicity, and bone marrow suppression, represent a major challenge for health care providers and have a significant impact on therapeutic decisions. In addition, cytotoxic agents impair the immune system, which has a critical role against cancers [9]. As a result, targeted delivery systems, such as drug-loaded liposomes and nanoparticles, have been developed for specific delivery to tumor cells with minimal systemic exposure.

The anthracycline drug class is one of the most active single cytotoxic agents in metastatic breast cancer. The major actions of anthracyclines are DNA intercalation, inhibition of topoisomerase II, and the formation of free radicals [10]. Daunorubicin (DNR) is a non-specific anthracycline antibiotic and has been used in treating a wide range of cancers, including breast cancer [11]. Clinical use of DNR is limited by two major problems, systemic toxicity, mainly cardiotoxicity, and drug resistance [12].

Liposomes are well recognized for drug and gene delivery with clinical evidence of efficacy. Liposomes are lipid vesicles that allow the delivery of hydrophilic molecules in the aqueous compartment and lipophilic molecules in the lipid bilayer [13]. Liposomes have several advantages for drug delivery of chemotherapeutic drugs. They have a role in delivering poorly soluble drugs, providing targeted drug delivery, reducing the toxic effect of drugs, extending circulation half-life, altering tissue distribution [14], being effective in overcoming multidrug resistance, and enhancing the therapeutic index [15].

Although PEGylation improves circulation half-life by reducing Reticuloendothelial System (RES) uptake, it decreases targeted liposomal accumulation and drug release by steric hindrance effects [16]. As a result, new liposome formulations where drug release is promoted in the vicinity of the tumor by physiological stimuli (such as pH) or physical external stimuli (such as heat) have been prepared. Thermosensitive liposomes (TSLs) are a distinctive class of triggerable liposomes as they promote drug release into the tumor vasculature and interstitial space under the influence of mild hyperthermia [17]. Inclusion of lipids with transition temperatures (40–45 °C) closer to physiological body temperature into liposomes allows generation of high intravascular drug concentration and a significant increase of anthracycline release into the tumor tissue after external localized heating [18]. Typically, TSLs consist of 1,2-dipalmitoyl-sn-glycero-3-phosphocholine (DPPC) (T_m_ = 41.4 °C) alone or with 1-myristoyl-2-stearoyl-sn-glycero-3-phosphocholine (MSPC) (T_m_ = 40 °C) [19]. TSLs are stable in circulating blood (at 37 °C); however, they release their contents rapidly when exposed to a mildly hyperthermic temperature (at 42 °C) because vesicles fuse and form planar bilayers, allowing rapid release of encapsulated molecules [20,21], hence increasing their concentration into the tumor tissue.

Cardiolipin (CL) differs from other glycerophospholipids because it is composed of four fatty acyl chains and three glycerol moieties, resulting in a negatively charged, cone-shaped structure [22]. CL plays a role in various cellular functions and signaling pathways inside and outside of mitochondria. It triggers apoptosis by inducing a structural defect of the inner mitochondrial membrane [23]. CL forms raft-like microdomains that may relocate and change lipid organization of both mitochondrial outer and inner membranes. Such transbilayer lipid movement eventually leads to membrane permeabilization and cell death [24]. Interestingly, CL also promotes extra-mitochondrial membrane permeabilization as it alters the mechanical stability of the membrane due to a decrease in lipid packing and formation of nonlamellar structures, resulting in the deformation of the biological membrane [25]. It has been proposed that CL plays an important role in the regulation of programmed cell death. CL permeabilization of the outer mitochondrial membrane initiates the release of apoptotic factors [26] because it allows specific targeting of *truncated Bid* (*tBid*) to the mitochondria [27] and facilitates its binding with *Bcl-xL* [28]. As a result, BAK and Bax, pro-apoptotic pore-forming proteins are activated, which are assumed to be responsible for the permeabilization of mitochondrial membranes. Many studies have determined that the incorporation of CL, besides inducing structural changes in the mitochondrial membrane, can also trigger structural changes in the cell membrane, making the membrane structurally deformed and more permeable [29]. In this study, we report a TSL liposomal formulation enriched with CL for enhanced cellular uptake and cytotoxicity in breast cancer cells.

## 2. Results and Discussion

### 2.1. Formulation Preparation

The liposomes prepared with CL were evaluated for EE and DL%. As shown in Table 1, the EE% and DL% for the formulations were above 90 and 15, respectively. DNR was entrapped inside liposomes by the remote loading method. This is one of the best approaches to attain a high EE%. Anthracycline drugs have been loaded into TSL successfully by this strategy [30]. The high EE% of amphipathic weak bases, such as DNR, is achieved by a transmembrane ammonium sulfate gradient across the liposomes bilayer (active loading) [31,32]. Similar to most drugs, DNR was not efficiently entrapped into the aqueous phase of the liposome without a pH gradient [33]. In remote loading, liposomes are initially prepared in an acidic environment, below their pKa. After vesicle self-assembly and dialysis, the core of the liposome remains acidic while the extravesicular pH level is similar to physiological conditions [32]. Remote loading of the uncharged drug allows molecules to diffuse across the bilayer of the liposomes to the aqueous interior where they become protonated. The positively charged drug can no longer cross the bilayer membrane and is trapped inside the liposomes [34].

Mole% PEG can significantly affect the EE%. An inverse relationship existed between mole% of PEG and EE of drugs since PEG might occupy some space in the core of the liposomes [35]. Mole% of PEG used in our formulation does not affect DNR EE%. There is no significant difference in EE% between liposomal formulations with PEG (F1) and F3 (without PEG) (*p* > 0.05). In addition, DNR:lipid at a ratio of 1:5 used in our TSL formulations obtained the desirable EE (above 90%). The drug-to-lipid ratio has a great influence on the EE% of anthracycline drugs such as DOX and DNR. The EE% decreases with increased anthracycline drug concentration [36,37].

### 2.2. Characterization of NP Formulations

Two types of TSLs, with and without CL, were fabricated via the film evaporation/extrusion method. As shown in Table 1, the particle size of F1 was about 115 nm with a polydispersity of 0.12, indicating uniform and dispersed liposomes. The particle size of F2 was about 123 nm with a polydispersity of 0.11. The zeta potential of F1 was approximately ~27 mV due to the presence of CL. CL is a quadruple-chained anionic amphiphile lipid composed of two 1,2-diacyl phosphatidate moieties esterified to the 1- and 3-hydroxyl groups of a single glycerol molecule. Under physiological conditions, phosphodiester moieties should both be negatively charged [38]. Moreover, PEG-DSPE lipid, incorporated into liposomes to extend the circulation time, imparts a negative charge [39]. The zeta-potential of F2 was less negative than F1 due to the absence of CL. The zeta-potential of F3 proved to be around neutral.

As shown in Table 1, there is no significant difference in particle size between TSL formulations prepared with or without CL (*p* > 0.05). Particle size is a very important parameter in the pharmacokinetics of the entrapped drug. Liposomes that have particles between 70–120 nm theoretically can pass through large fenestrations, sinusoidal capillaries as an example, and at the same time provide a greater carrying capacity than smaller liposomes [40]. Liposomes with diameters larger than 200 nm have high RES uptake [41] while smaller particles diminish the uptake by RES and are preferred for tumor targeting [16]. The liposome preparation of F1 allowed reproducible liposome formation with diameters below 100, small PI (<0.2) and high entrapment efficiency (>90%).

### 2.3. Temperature Triggered Release In Vitro

The release of DNR from different liposomal formulations at 37 °C and 42 °C is depicted in Figure 1. As expected, the release of DNR was very slow (no more than 5% DNR at one hour) from both F1 and F2 TSLs at 37 °C. However, the release of DNR was significantly accelerated when the temperature was above the phase transition temperatures of DPPC and MSPC (42 °C). The cumulative release of DNR reached 90% within one hour. Regarding the non-TSL formulation (F3), temperature had almost no effect on the release of DNR.

The development of drug delivery nanocarriers with controllable drug retention and release characteristics is a challenge. The therapeutic efficacy of regular liposomal anthracycline is not enhanced considerably because of inadequate drug release at the site of action. However, long-circulation times and reduction of drug-associated toxicity were observed [42,43]. Typically, liposomes release their contents by passive diffusion or liposome degradation, which is not favorable for non-cell cycle specific anthracycline drugs because the therapeutic concentration might not be achieved [44]. Thus, to optimize the amount of drug release, liposomal systems that release drugs at the target site in response to a specific stimulus, such as pH or mild hyperthermia, have been developed.

The first TSL formulation was prepared using DPPC and DSPC lipids [45]. Since then, TSLs have been further developed. The idea is that liposomes have lipids that undergo phase transitions in response to heating [46]. Below their transition temperature, lipids exist in gel phase and are well ordered and packed; however, when temperature is elevated and approaches their transition temperature, the mobility of the lipid increases and liposome bilayers change from a solid gel phase to a liquid crystalline phase [47]. This phase transition makes the membrane more permeable to water and encapsulated drugs [47,48]. DPPC lipid (T_m_ = 41.4 °C) is a key component in most TSL formulations and is usually used with other lipids such as MSPC (T_m_ = 40 °C) or MPPC (T_m_ = 44 °C), and DSPE-PEG to increase the circulation time [49]. Several TSLs have been designed to enhance anthracycline drugs release in response to mild hyperthermia. A liposomal formulation composed of DPPC/HSPC/cholesterol/DPPE-PEG 50:25:15:3 (mol/mol) exhibited fast release of DOX (60% released within 30 min) when incubated at 42 °C [50]. Another lysolecithin-containing thermosensitive formulation composed of DPPC:MPPC:DSPE-PEG-2000 in the molar ratio of 90:10:4 accelerated the release rate of DOX from liposomes at 41.3 °C (80% of DOX released in 20 s) [51].

TSL formulations in our study exhibited a slow drug-release profile at 37 °C. There was no significant difference in the release profile between F1 and F2, indicating that the addition of CL has no effect on drug release (*p* > 0.05). Under the influence of mild hyperthermia, both F1 and F2 released DNR completely within an hour. There was no significant difference in the release profile between F1 and F2, indicating that the addition of CL did not slow DNR release. Extended exposure to mild hyperthermia is not necessary. After decreasing the temperature below the transition temperature, the membrane will not solidify homogenously since solid domains will be formed within the membrane [52].

### 2.4. Cytotoxicity of DNR-Loaded TSLs

From the dose-response curves for cells incubated with various formulations for 48 h, the IC50 values were determined. The IC50 values for free DNR, F3, F2 and F1 are 1.9 ± 0.15 μM, 2.1 ± 0.4 μM, 2 μM ± 0.31 and 0.5 μM ± 0.15, respectively (Figure 2A). Thus, the IC50 of F1 was about 3.8-fold lower than DNR solution. Furthermore, the IC50 F1 was about 4-fold lower than F2.

Anthracycline DNR is an antineoplastic drug that exhibits antitumor activity against a wide variety of cancers including breast cancer cells and cancer stem cells [53]. However, its use is limited by the development of hematotoxicity, nephrotoxicity, peripheral neuropathy, and cardiomyopathy [54]. Encapsulating chemotherapy drugs in a biocompatible material that can deliver them to their intended site of action might provide a solution to this problem [55]. Liposomes are considered one of the most successful drug delivery systems applying nanotechnology to increase circulation time and reduce toxicities of conventional drugs by a change in drug distribution in the body [56]. Despite the fast release and targeting the tumor site, the efficacy of therapeutic molecules is often limited by the insufficient accumulation in target tissues [57].

Herein, we report the development of a CL thermosensitive liposomal formulation composed of DPPC/MSPC/DSPE-mPEG (2000)/CL, to enhance both the release and uptake of DNR against MDA-MB-231 breast cancer cells. Our in vitro cytotoxic data provided clear evidence to support DNR TSLs with CL (F1) as more effective in breast cancer cell growth inhibition compared with TSLs without CL (F2) and free DNR (Figure 2B).

We did not observe any significant cell damage to MDA-MB-231 when 10 min mild hyperthermia was applied. 10 min was enough to release DNR. A positive impact on the amount and rate of DNR released in the presence of multiple types of lipids resulted from an increase in packing incompatibility and, hence, increased permeability when mild hyperthermia was applied [58]. Furthermore, we did not observe any significant enhancement in cellular toxicity between free DNR and F2, indicating complete drug release from TSLs. In addition, empty liposome formulation containing CL (F4) did not exhibit any effect on cell survival. Thus, the increased efficacy of F1 might contribute to enhancing intracellular DNR delivery.

This is a strong indication that CL lipid affected the membrane permeability since anthracycline drugs enter cells by passive diffusion [59], and the anti-tumor effect is enhanced by increasing cellular uptake [60]. CL can form domains on the cell membrane that ultimately change membrane physical stability allowing more drugs to enter into cells [25].

Besides triggering drug release from the TSLs, mild hyperthermia can increase tumor vascular permeability and, thus, extravasation and accumulation of liposomes in the tumor site [17]. We conducted cell cytotoxicity (MTT) assay using a high concentration of DNR without exposure of cells to mild hyperthermia to assure that TSLs will not release DNR unless exposed to mild hyperthermia and to assure that CL has no cytotoxicity at 37 °C. As shown in Figure 3, after a prolonged incubation time (24 and 48 h) at 37 °C, no significant reduction in cell viability (due to cytotoxicity) was observed from TSLs due to lack of drug release. Our TSLs enriched with CL were stable in physiological environments; however, they were versatile by modulation of temperature as intended.

### 2.5. DNR Accumulation into MDA-MB-231 Cell Lines

The DNR uptake by tumor cells treated with different liposome formulations, all containing 14 μM DNR, were evaluated at 37 °C and 42 °C. As shown in Figure 4A, after 0.5 h at 37 °C free DNR showed the highest uptake and TSL uptake showed the lowest. However, after 4 h, F3 showed a higher uptake compared with free DNR while TSLs still did not exhibit any significant uptake. In Figure 4B, mild hyperthermia enhanced DNR uptake significantly, especially at 4 h. When treated with F1 for 4 h, DNR accumulated at 240%, 256% and 152% compared to F2, free DNR, and F3, respectively.

PEGylation of liposomes significantly reduced cellular uptake and drug release of the liposomes, which interferes with the antitumor efficacy of liposome-encapsulated drugs [61]. However, PEGylation is still very important as it increases liposomes circulation time by decreasing opsonization and RES uptake. Liposomes without PEG are taken easily by endocytic process and their uptake is time and concentration-dependent [62]. Several stimuli-responsive liposomes, such as light, enzyme, pH, and temperature-sensitive liposomes, have been formulated to increase drug release from liposomes at tumor site.

At 37 °C, free DNR exhibited the highest cellular uptake in early-stage (0.5 h) because drug solution could be easily diffused through the lipid bilayer of cells; however, after 4 h, F3 exhibited more intracellular accumulation of DNR because this formulation was not grafted with PEG and can be taken more effectively by endocytosis. TSLs either with or without CL had good stability and did not release DNR. When applying mild hyperthermia, rapid release of DNR was observed from our TSLs formulation. At 4 h, we observed the enhanced effect of CL on DNR accumulations as more DNR accumulated within cells in the case of F1 compared to other formulations. Enrichment of TSLs with CL enhanced the cellular uptake; this might be due to the ability of CL to form domains on the cell membrane that eventually change its physical stability, fluidity, and mechanical structure, allowing more drug to accumulate into cells [24,29]. Cellular uptake results are consistent with both release and cytotoxicity results.

Cardiolipin (CL) is a negatively charged lipid [22], which can induce lipid structural defects in the cell membrane and mitochondrial membrane [23]. CL can integrate with the cellular lipid domains and induce structural changes in the cell membrane, making the membrane structurally deformed and more permeable [29]. Therefore, in the current study we find that the DNR uptake in the cells by the F1 formulation (with CL) is significantly higher compared to F2 (no CL) and F3 (formulation similar to DaunoXome).

### 2.6. DNR Internalized Efficiently from CL-TSLs in MDA-MB-231 Cancer Cells

Confocal fluorescence microscopy was used to observe the intracellular uptake of DNR from TSLs and the standard DNR at 37 °C and after exposure to mild hyperthermia. Intracellular incorporation of DNR in MDA-MB-231 revealed no DNR uptake at 37 °C for F1 and F2 compared with free DNR and F3 (Figure 5). After exposing cells containing F1 and F2 to mild hyperthermia, a significant and visible increase in DNR uptake was observed (Figure 6).

F1 displayed a significantly higher DNR accumulation in a time-dependent manner. After 6 h of incubation (14 μM DNR), the fluorescence levels were consistently higher in F1 compared with F3, F2, and free DNR. CL changes the physical properties of cell membranes such as thickness and permeability leading to more DNR accumulation inside cells. To examine the interaction of CL with the cellular membrane, we incorporated fluorescent CL into the liposomal formulation encapsulated with DRN. Furthermore, we made a blank liposomal formulation with fluorescent CL. As shown in Figure 7, CL interacted with the cell membrane leading to more DNR accumulation inside cancer cells.

### 2.7. Short-Term Stability Studies

The particle size, EE%, DL%, PI, zeta potential, and osmolarity of different formulations kept at 4 °C were monitored for one month as shown in Table 2. There were no significant changes for any of the indexes, including EE%, suggesting that there was no significant drug leaked from TSLs. Slow leaking and high stability always characterized the most promising liposomal system with desirable efficacy. Stability data indicated that CL did not destabilize the liposomal membrane. Since anthracycline drugs precipitate as fibrous-bundle aggregates in liposomes [63], high drug: lipid ratio might cause liposomal deformation. The drug: lipid ratio (1:5) used in our formulations does not cause liposomal membrane deformation, which explains the good stability profile, especially the EE%. In addition, the most stable liposomal formulation was obtained by incorporation of 30 mol% cholesterol [64].

## 3. Materials and Methods

### 3.1. Materials

1,2-dipalmitoyl-*sn*-glycero-3-phosphocholine (DPPC), 1-myristoyl-2-stearoyl-*sn*-glycero-3-phosphocholine (MSPC), 1,2-distearoyl-*sn*-glycero-3-phosphoethanolamine-N-[methoxy(polyethylene glycol)-2000] (ammonium salt) (DSPE-mPEG (2000)), cardiolipin (CL), 1,1′,2,2′-tetraoleoyl cardiolipin [4-(dipyrrometheneboron difluoride)butanoyl] (ammonium salt), Distearoylphosphatidylcholine (DSPC), TopFluor^®^ CL were purchased from Avanti Polar Lipids Inc. (Alabaster, AL, USA). Cholesterol and ammonium sulfate were purchased from JT Baker (Phillipsburg, NJ, USA). Fetal bovine serum (FBS), Dulbecco’s Modified Eagle’s Medium (DMEM) and other reagents for cell culture were purchased from Mediatech (Manassas, VA, USA). Daunorubicin was purchased from AvaChem Scientific (San Antonio, TX, USA). 3-(4,5-Dimethyl-2-thiazolyl)-2,5-diphenyl-2H-tetrazolium bromide (MTT) assay kit and Phosphate Buffered Saline (PBS) was purchased from Sigma-Aldrich (St. Louis, MO, USA). DAPI (4′,6-diamidino-2-phenylindole) and Bicinchoninic acid protein kit was purchased from Thermo Fisher Scientific (Rockford, IL, USA). Polycarbonate membrane (0.08 μm) was purchased from Whatman plc (Maidstone, Kent, UK). MDA-MB-231 breast cancer cells were obtained from American Type Culture Collection (Manassas, VA, USA).

### 3.2. Liposomes Preparation

Liposomes were prepared by lipid thin-film hydration technique using rotary vacuum evaporator. Briefly, DPPC, MSPC, cholesterol, DSPE-mPEG (2000) and CL were prepared as 10 mg/mL solution individually in chloroform. These solutions were mixed at a molar ratio of 57:40:30:3:20 for DPPC/MSPC/cholesterol/DSPE-mPEG (2000)/and CL, respectively. The mixture was evaporated on a rotavapor (Rotavapor, Büchi, Germany) by applying a vacuum of about 25 mmHg at 65 °C, until it formed a thin film in the flask. The lipid film was further dried under a stream of nitrogen for 1h, followed by vacuum desiccation for 2 h. The dry lipid film was then hydrated in 250 mM ammonium sulfate solution (pH 5.5). This mixture was then placed in a water-bath incubator (65 °C) for 1 h (with intermittent vortex) to form coarse liposomes and underwent seven liquid nitrogen freeze–thaw cycles above the phase transition temperature of the primary lipid. The liposome mixture was then extruded (10 passes) through 80 nm polycarbonate filter using Lipex^®^ 10 mL barrel extruder (Transferra Nanosciences Inc., Burnaby, BC, Canada). The free ammonium sulfate outside the liposomes was removed by dialysis (regenerated cellulose, 12,000 to 14,000 Daltons molecular weight cut off dialysis tubing) against sucrose solution (10% *w*/*v*, 250 mL) at 4 °C. An isotonic sucrose solution (10% *w*/*v*) was discarded and replaced with a fresh solution after 1, 4, 8 h intervals and then left overnight. The total phospholipid levels of each formulation were measured by a colorimetric assay that measured inorganic phosphate after acidic digestion [65]. Liposomal formulation similar to DaunoXome^®^, composed of DSPC/cholesterol/DNR (in a 10:5:1 molar ratio), was prepared by the same method; however, citrate was used instead of ammonium sulfate to hydrate the lipid film. Table 3 summarizes the different formulations prepared.

### 3.3. Drug Encapsulation in Liposomes (Active Loading)

DNR solution of an appropriate concentration was prepared by adding the required quantities of the drug in PBS. This solution, after adjusting the pH to 8, was added to the liposomes at appropriate drug-to-lipid ratios (0.2:1). Excess DNR was then removed by sequential dialysis against sucrose solution (10%) at 4 °C. Based on initial results of drug loading efficiency, 1:5 drug-to-lipid ratio was found to be optimum and was used for all formulations.

### 3.4. Encapsulation Efficiency (EE%) and Drug Loading (DL%) Measurement

The amount of DNR entrapped into liposomes (EE% and DL%) was determined fluorometrically at 480 nm (excitation) and 590 nm (emission) using a microplate reader 142 (Fluostar, BMG Labtechnologies, Germany). Briefly, Triton X-100 (1%) was added to different liposomal DNR to break the liposome bilayer and release the entrapped DNR. Liposomal drug concentration was calculated from the DNR standard curve. All experiments were run in triplicate and mean data were presented.

The EE% was calculated as follows:$$\mathrm{Encapsulation}\;\mathrm{Efficiency}\;\left(\%\right)=\frac{\frac{\left(\mathrm{amount}\;\mathrm{of}\;\mathrm{liposomal}\;\mathrm{drug}\right)}{final}lipid}{\mathrm{total}\;\mathrm{amount}\;\mathrm{of}\frac{\mathrm{drug}}{\mathrm{initial}\;\mathrm{lipid}}}\times 100$$

The DL% was calculated as follows:$$\mathrm{Drug}\;\mathrm{Loading}\;\left(\%\right)=\frac{\left(\mathrm{amount}\;\mathrm{of}\;\mathrm{liposomal}\;\mathrm{drug}\right)}{\mathrm{total}\;\mathrm{amount}\;\mathrm{of}\;\mathrm{drug}\;\mathrm{added}\;+\;\mathrm{amount}\;\mathrm{of}\;\mathrm{excipients}\;\mathrm{added}}\times 100$$

### 3.5. Determination of Particle Size of Liposomal Formulations

The particle size distribution of the liposomal formulations was studied by the dynamic light scattering method using Nicomp 380 ZLS particle size analyzer (Particle Sizing Systems, Santa Barbara, CA, USA). Mean particle size and polydispersity index of the formulations after appropriate dilutions were calculated. All determinations were performed in triplicate at room temperature (25 °C). The samples were diluted about 100 times.

### 3.6. Determination of Zeta Potential of Liposomal Formulations

Measurements of liposome zeta potential were carried out by photon correlation spectroscopy (PCS, Zetatrac, Largo, FL, USA). For the analyses, formulations were diluted in an aqueous medium. All determinations were performed in triplicate at room temperature (25 °C).

### 3.7. Determination of Osmolality of Liposomal Formulations

Osmolarity of the formulations was analyzed by a vapor pressure osmometer (model K-7000 Knauer, Berlin, Germany). Before performing the analyses, the osmometer was calibrated with a solution of NaCl (400 mOsm). The determinations were made in triplicate at 25 °C.

### 3.8. In Vitro Release Studies

The release profile of DNR from liposome formulations was determined by the dialysis method. PBS (250 mL, pH 7.4) in 250 mL conical flasks was used as a receptor phase. Regenerated dialysis tubing (12,000 to 14,000 Daltons molecular weight cut off), 30 mm × 25 mm release area, pre-soaked in buffer solution for one hour, was used. 1 mL of the formulation or DNR solution was placed in the dialysis tubing while immersed in the receptor phase. All flasks were incubated at 37 °C or 42 °C in a rotary shaker set at 150 rpm. Samples (1 mL) were collected at different time intervals and the sample volumes were replenished with fresh buffer immediately. The concentration of DNR in the receptor buffer (dialysate) was analyzed fluorometrically (480 nm excitation and 590 nm emission) using a microplate reader. The cumulative amount of DNR released versus time was plotted. All experiments were run in triplicate and mean data were presented.

### 3.9. Stability Studies

Short-term physical stability was on the liposomes. All liposomal formulations were stored at 4 °C under N_2_ and protected from light for one month and particle size, polydispersity, zeta potential, and osmolarity were then determined.

### 3.10. Cell Culture

MDA-MB-231 cells were cultured in Dulbecco’s Modified Eagle’s Medium (DMEM). The medium was supplemented with 10% (*v*/*v*) fetal bovine serum (FBS), 100 U/mL penicillin, and 100 μg/mL streptomycin at 37 °C in a humidified atmosphere containing 5% CO_2_. All experiments were performed at a confluence of 90 to 95%.

#### 3.10.1. Measurement of Cytotoxicity by MTT Assay

MDA-MB-231 cells were cultured in flat-bottom 96-well plates for 24 h. The cell density in the wells was around 8 × 10^3^ cells/well. The cells received treatments of various liposomal formulations (0.01 μM, 0.05 μM, 0.1 μM, 0.5 μM, 1 μM, 2 μM and 3 μM) for 48 h prior to MTT assay. TSL treated cells were heated by placing the well plates in a precision-controlled incubator at 42 °C for 10 min (after the incubator had reached thermal equilibrium), then returned back to 37 °C or only incubated at 37 °C. After the treatments, 10 μL of 3-[4, 5-dimethylthiazol-2-yl]-2, 5-diphenyl tetrazolium bromide (MTT) was added to each well and the cells were incubated at 37 °C for an additional 2 h. Finally, the medium was aspirated and 200 μL dimethylsulfoxide (DMSO) was added to each well to solubilize the dye remaining in the plates. The absorbance was measured using a microplate reader (Spectramax M5, molecular devices, Sunnyvale, CA, USA) at 544 nm. All experiments were run in triplicate and mean data were presented.

#### 3.10.2. Cellular Daunorubicin Uptake

MDA-MB-231 cells were cultured in flat-bottom 24-well plates. At confluence, cells were exposed to 14 μM of liposomal DNR or free DNR for 0.5 and 4 h at different temperature (37 °C and 42 °C). TSL treated cells were heated by placing the well plates in precision-controlled incubator at 42 °C for 10 min (after the incubator had reached thermal equilibrium), then returned to 37 °C. After extensive washing with PBS, cells were lysed in 100 μL of 1% Triton X-100. DNR fluorescence was measured by a microplate reader at 480- and 590-nm for excitation and emission, respectively. After calculating cellular DNR contents with a standard curve, all contents were corrected for any differences in protein content using the bicinchoninic acid assay [66]. In addition, all values were corrected for background fluorescence. All the experiments were run in triplicate and mean data were presented.

#### 3.10.3. Fluorescence Microscopy

MDA-MB-231 cells were seeded in a flat-bottom 24-well plate for 24 h. After exposure to liposomal DNR or free DNR 14 μM for 0.5 and 6 h at 37 °C or 42 °C (TSL treated cells were heated at 42 °C for 10 min then returned to 37 °C), cells were washed and fixed (15 min in 4% (*w*/*v*) paraformaldehyde in phosphate-buffered saline). All samples were examined with a fluorescence microscope (EVOS fl, ZP-PKGA-0494 REV A, Bothell, WA USA) and photographed through at 20× magnification.

### 3.11. Statistical Analysis

The DNR% released from liposomes was plotted as a function of time (h). All data were presented as mean ± standard deviation. GraphPad Prism software was used to determine the standard deviation and statistical levels of significance. All data were subjected to one-way analysis of variance (ANOVA) to determine the statistical levels of significance. *p*-value less than 0.05 was considered statistically significant

## 4. Conclusions

We have designed TSLs using DPPC, MSPC, DSPE-mPEG, and cholesterol. The cytotoxicity and cellular uptake of TSLs were further increased by the inclusion of CL. The characteristics of selected formulations were investigated in vitro for drug release at different temperatures, DNR accumulation, and antitumor efficacy. The results demonstrated that our TSLs with CL released their contents rapidly when exposed to mild hyperthermia for a short time. Moreover, our liposomal system was confirmed to be highly stable in physiological environments and during storage. CL enriched liposomes exhibited a higher cytotoxic and cellular uptake by MDA-MB-231 cell lines than the same formulation without CL, free DNR, and liposomes similar to DaunoXome^®^. Therefore, this formulation appears to be a promising delivery system in the treatment of breast cancer with the potential to provide site-specific chemotherapy, with higher therapeutic efficiency due to its enhanced permeability while causing less adverse effects.

## Figures and Tables

**Figure 1 ijms-23-11763-f001:**
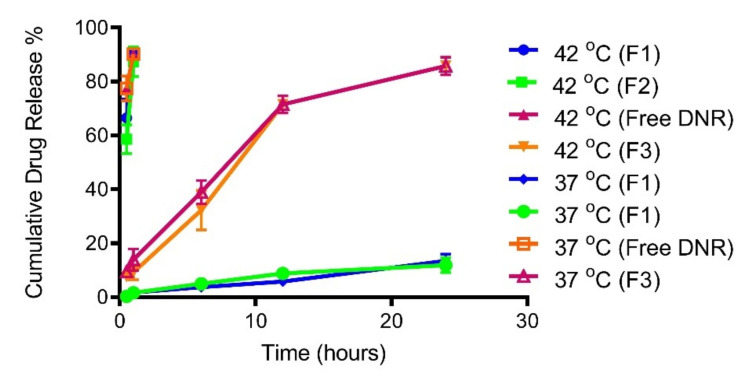
Release profiles of DNR from TSL and non-TSL formulations at 37 °C and 42 °C. Non-TSLs were not influenced by temperature. Values represent mean ± standard deviation (*n* = 3).

**Figure 2 ijms-23-11763-f002:**
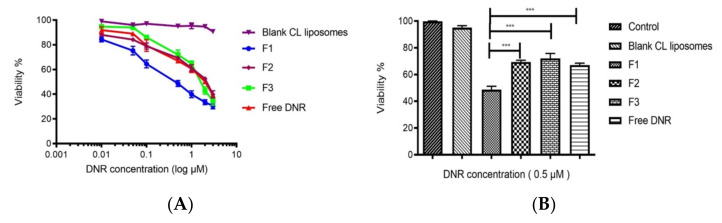
(**A**): CL potentiates the cytotoxic effect of DNR TSLs against MDA-MB-231 cell lines. Cells were incubated at 42 °C for 10 min then returned to 37 °C. (**B**): In vitro cytotoxicity of different formulations against MDA-MB-231 breast cancer cells. Cells were incubated at 42 °C for 10 min then returned to 37 °C for 48 h. *** *p* < 0.001. Values represent mean ± standard deviation (*n* = 3).

**Figure 3 ijms-23-11763-f003:**
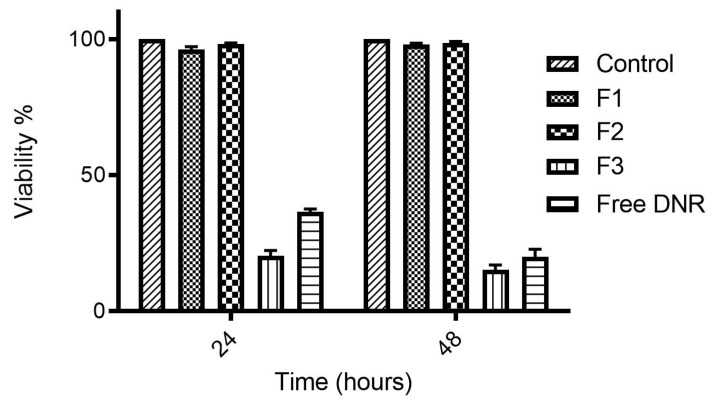
Cytotoxic activity (MTT assay) of different types of TSLs (F1 and F2) compared to F3 and free DNR. The cytotoxic activity of DNR loaded TSLs was studied at 14 μM DNR at 37 °C on MDA-MB-231cell lines. Values represent mean ± standard deviation (*n* = 3).

**Figure 4 ijms-23-11763-f004:**
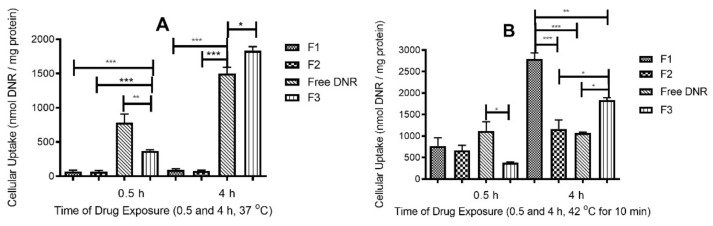
(**A**): Cellular uptake of TSLs DNR (14 μM) at 37 °C by MDA-MB-231 cancer cell lines. * indicates *p* < 0.05, ** indicates *p* < 0.01, and *** indicates *p* < 0.001. Values represent mean ± standard deviation (*n* = 3). (**B**): Effect of TSLs enriched with CL on DNR uptake by MDA-MB-231 cancer cell lines after exposure to mild hyperthermia. * *p* < 0.05, ** *p* < 0.01, and *** *p* < 0.001. Values represent mean ± standard deviation (*n* = 3).

**Figure 5 ijms-23-11763-f005:**
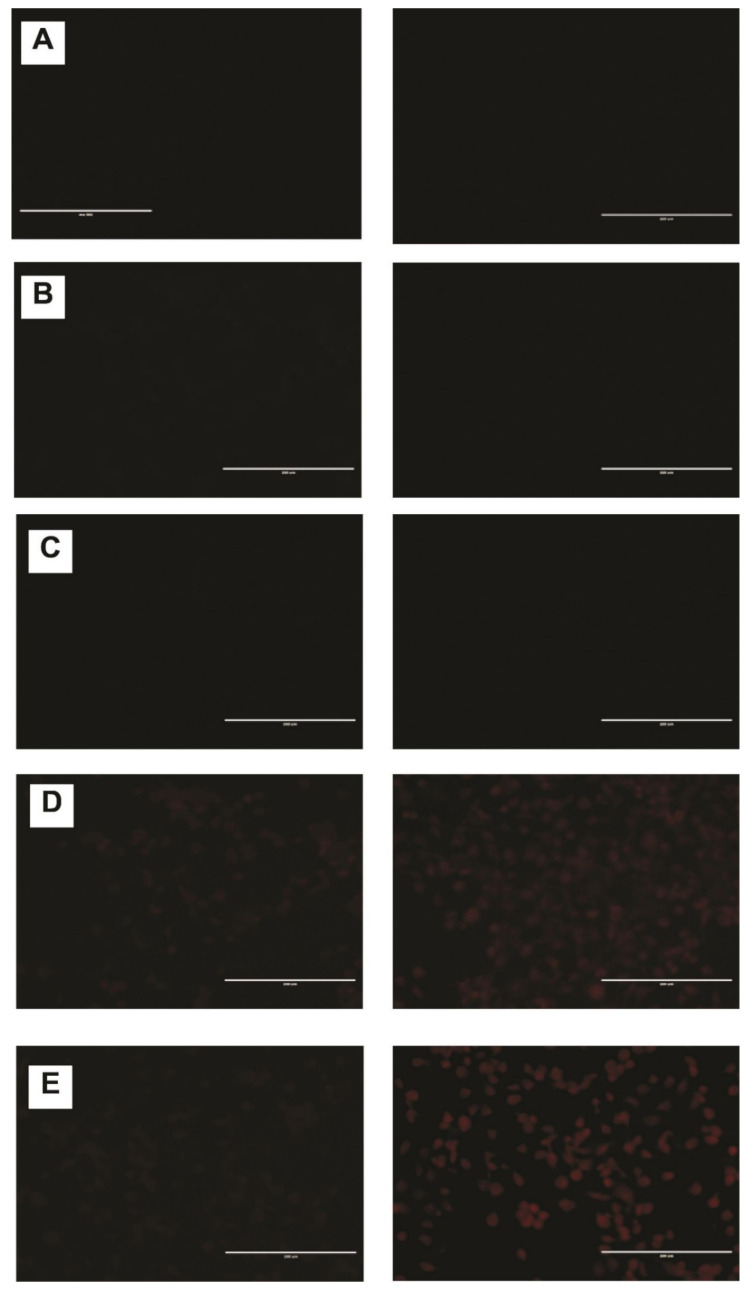
Fluorescence microscopy of different liposomes at 37 °C after 0.5 h (**left**) and 6 h (**right**). MDA-MB-231 cells were cultured for 24 h (**A**) and then were treated with F1 (**B**) F2 (**C**), free DNR (**D**) or F3 (**E**). Final liposomal DNR concentrations were 14 µM. Scale bar = 200 µm. Where there is no drug accumulation, the images appear darker with no cell line impressions. TSLs-CL formulation with red color shows DNR accumulation.

**Figure 6 ijms-23-11763-f006:**
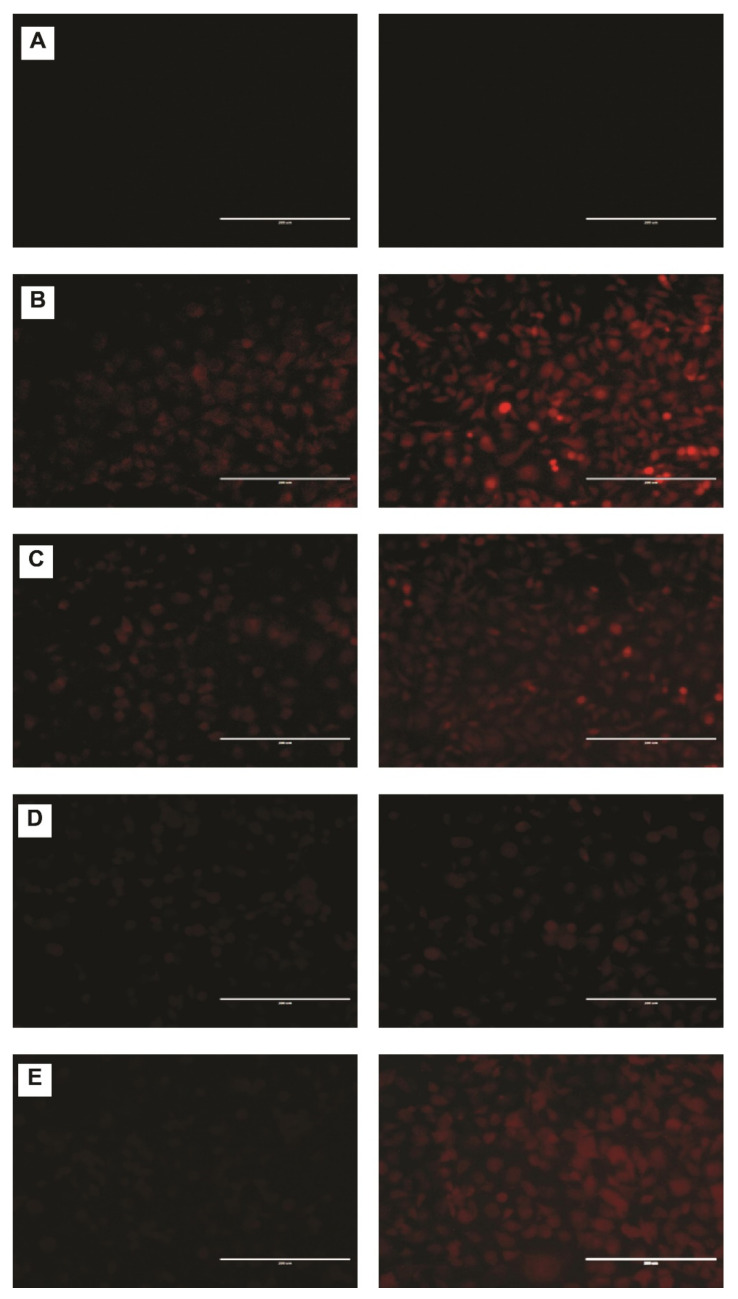
Fluorescence microscopy showing CL enhanced DNR uptake from TSL after exposure to mild hyperthermia (42 °C, 10 min) after 0.5 h (left) and 6 h (right). MDA-MB-231 cells were cultured for 24 h (**A**) and then were treated with F1 (**B**), F2 (**C**), free DNR (**D**) or F3 (**E**). Final liposomal DNR concentrations were 14 µM. Scale bar = 200 µm. Where there is no drug accumulation, the images appear darker with no cell line impressions. TSLs-CL formulation with red color shows DNR accumulation.

**Figure 7 ijms-23-11763-f007:**
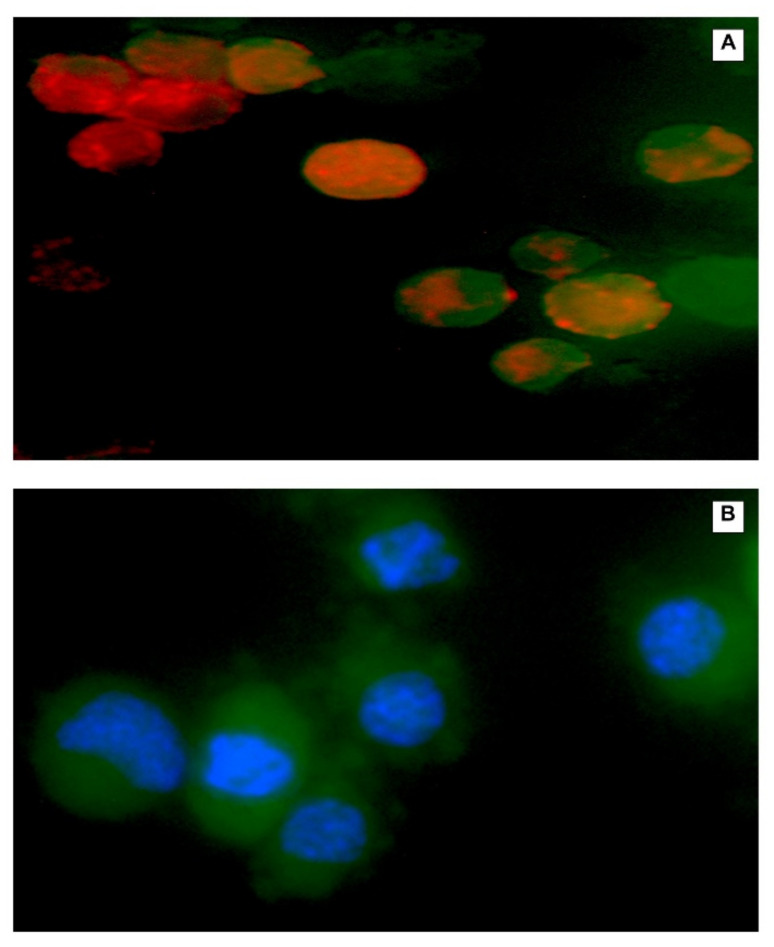
Fluorescence microscopy showing CL interacting with the cellular membrane. Fluorescent CL liposomal formulation with DNR (**A**), and fluorescent CL liposomal formulation with DAPI staining (**B**). Red color represents uptake of doxorubicin in cells. Blue color represents nuclei of cells stained with DAPI while green color represents fluorescent CL entry into cells.

**Table 1 ijms-23-11763-t001:** Physicochemical characteristics of different liposome formulations. Values are expressed as mean ± SD, *n* = 3.

Formulation	EE %	DL (%)	Particle Size (nm)	PI	Zeta Potential (mV)	Osmolality (Osm/L)
F1	98.0 ± 0.8	16.2 ± 0.4	115 ± 1.3	0.12 ± 0.03	−27.7 ± 1.9	309 ± 3.6
F2	96.0 ± 3.6	15.7 ± 0.7	124 ± 1.7	0.11 ± 0.01	−9.7 ± 2.8	300 ± 14.9
F3	94.0 ± 3.7	17.0 ± 0.9	85.0 ± 3.4	0.15 ± 0.03	−2.5 ± 2.1	280 ± 14.5

EE = Encapsulation Efficiency; DL = Drug Loading; PI = Polydispersity Index.

**Table 2 ijms-23-11763-t002:** One month stability at 4 °C under N_2_ and protected from light. Values represented as mean ± SD, *n* = 3.

Time Points	Initial Analysis			After 1 Month Analysis		
Formulation	F1	F2	F3	F1	F2	F3
Particle Size (nm)	115 ± 1.3	124 ± 1.7	85.0 ± 3.4	119 ± 2.5	125 ± 1.4	88.0 ± 2.4
PI	0.12 ± 0.03	0.11 ± 0.01	0.15 ± 0.03	0.22 ±0.01	0.17 ± 0.06	0.16 ± 0.07
EE%	98.0 ± 0.8	96.0 ± 3.6	94.0 ± 3.7	95.0 ± 2.5	93.0 ± 1.4	92.0 ± 2.1
DL%	16.2 ± 0.4	15.7 ± 0.7	17.0 ± 0.9	15.8 ± 0.4	15.4 ± 0.2	15.9 ± 0.7
Zeta Potential (mV)	−27.7 ± 1.9	−9.7 ± 2.8	−2.5 ± 2.1	−24.4 ± 1.4	−7.4 ± 2.9	−4.0 ± 2.1
Osmolarity (Osm/L)	309 ± 3.6	300 ± 14.9	280 ± 14.5	311 ± 6.7	314 ± 2.9	249 ± 11.2

I = Polydispersity Index; EE% = Encapsulation Efficiency; DL% = Drug Loading.

**Table 3 ijms-23-11763-t003:** Composition of various thermosensitive liposomal formulations. The quantities of lipids are expressed in molar ratios (with no units).

Ingredients	F1	F2	F3 *	F4 #
DPPC	57	57	-	57
MSPC	40	40	-	40
DSPC	-	-	10	-
Cholesterol	30	30	5	30
DSPE-mPEG (2000)	3	3	-	3
CL	20	-	-	20
DNR	1	1	1	-
DNR: lipid ratio	1:5	1:5	1:10:5	-

* Liposomal formulation similar to DaunoXome^®^. 1:10:5 is the molar ratio of daunorubicin:DSPC:cholesterol in liposomal formulation similar to DaunoXome^®^. # Blank formulation, used in the cytotoxicity studies.

## Data Availability

Not applicable.

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
