# Peer review of "Cardiolipin for Enhanced Cellular Uptake and Cytotoxicity of Thermosensitive Liposome-Encapsulated Daunorubicin toward Breast Cancer Cell Lines"

_ijms, 2022, doi:10.3390/ijms231911763_

Round 1

Reviewer 1 Report

1. The selection of phospholipid composition for thermosensitive property is relatively new. It is not clear how the liposomes are taken up by the cells. Please provide justification for higher cellular uptake of these liposomes.

2. How are the drug loaded liposomes were purified from the free drug? Is it by size exclusion chromatography (SEC)? Please include the protocol for SEC.

3. Line 407 and Line : Formatting is needed

4. Please discuss the scope and potential utility of the formulation in future.

Author Response

1. The selection of phospholipid composition for thermosensitive property is relatively new. It is not clear how the liposomes are taken up by the cells. Please provide justification for higher cellular uptake of these liposomes.

RESPONSE:

Authors appreciate reviewer’s feedback and comment. In the revised manuscript, the justification for higher cellular uptake has been added under section 2.5 (DNR accumulation) lines 316-321.

  1. How are the drug loaded liposomes were purified from the free drug? Is it by size exclusion chromatography (SEC)? Please include the protocol for SEC.

RESPONSE:

The authors appreciate the reviewer’s comment. We did not perform SEC for the removal of free drug. The excess DNR was removed by sequential dialysis against sucrose solution (10%) at 4oC. The cold temperature of the dialysis method ensures no entrapped drug is leached out from the liposomes. This information has been updated in the revised manuscript under section 3.3.

  1. Line 407: Formatting is needed

RESPONSE:

The authors appreciate the reviewer’s comment. Line 407 changed to:

The total phospholipid levels of each formulation were measured by a colorimetric assay that measured inorganic phosphate after acidic digestion

  1. Please discuss the scope and potential utility of the formulation in future.

RESPONSE:

The authors appreciate the reviewer’s feedback. We now added lines (540-542): “Therefore, this formulation appears to be a promising delivery system in the treatment of breast cancer with the potential to provide site-specific chemotherapy, higher therapeutic efficiency due to their enhanced permeability while causing less adverse effects.”

Reviewer 2 Report

The manuscript reports an interesting application of cardiolipin in thermosensitive liposomes encapsulating daunorubicin.

Some comments are:

Introduction:

Please define the abbreaviation RES, DPPC and MSPC (line 73 and Line 83).

Please discuss better the importance of incorporating cardiolipin into liposomes for drug delivery

Materiala and Methods

Improve caption of Table 3 and define the unit of the number in the Table 3. Table is erroneously indicated as Table 1 in the caption.

Paragraph 3.5 At which temperature particle size analysis was performed? How much samples were diluted?

Paragraph 3.8 How much volume of PBS medium was employed for the release study?

How was the cumulative percentage of drug release calculated?

Results and Discussion

Why results from F4 formulation (control) are not reported in Table 1?

Line 150 Table 1 is erroneusly referred as Table 2.

The choice of symbols in the graph and denomination of samples in the legend is confusing. I suggest changing the criteria about chosing the symbols for the plot and referring samples as F1, F2 and F3 formulations.

Paragaph 2.4 Line 242-255  The provided explanation about the reason of higher cytotoxicity for formulation containing cardiolipin is not clear.

The authors should dimostrate that cardiolipin form microdomain that can alter the organization of the phospholipid bilayer in liposomes, therefore incrreasing the cellular uptake and therapeutic outcomes. Alternatively, a sounded discussion between the obtained results and the literature available regarding liposomes could be useful.  

Conclusions:

Line 510 It seems from the sentence that the thermosensitive behaviour is related to cardiolipin and not phospholipid compositions. Instead, cardiolipin does not affect the thermosensitive behaviour of liposomes.

Author Response

The manuscript reports an interesting application of cardiolipin in thermosensitive liposomes encapsulating daunorubicin. RESPONSE: Authors thank the reviewer for their positive feedback.

Some comments are:

Introduction:

  1. Please define the abbreviations RES, DPPC and MSPC (line 73 and Line 83).

RESPONSE:

The authors appreciate the reviewer’s comment. Abbreviations have been defined in the entire revised manuscript.

  1. Please discuss better the importance of incorporating cardiolipin into liposomes for drug delivery

RESPONSE:

The authors appreciate the reviewer’s comment. We have the following information modified in the revised submission:

Introduction:

Many studies have determined that the incorporation of CL, besides inducing structural changes in the mitochondrial membrane, it also can trigger structural changes in the cell membrane, making the membrane structurally deformed and more permeable [29]. In this study we report, a TSL liposomal formulation enriched with CL for enhanced cellular uptake and cytotoxicity in breast cancer cells.

Results and Discussion (Section 2.5, based on both reviewers 1 and 2 comments)

Cardiolipin (CL) is a negatively charged lipid [22], which can induce lipid structural defects in the cell and mitochondrial membranes [23]. CL can integrate with the cellular lipid domains and induce structural changes in the cell membrane, making the membrane structurally deformed and more permeable [29]. Therefore, in the current study we find that the DNR uptake in the cells by the F1 is significantly higher compared to F2 (no CL) and F3 (formulation similar to DaunoXome). 

Materials and Methods:

1. Improve caption of Table 3 and define the unit of the number in the Table 3. The table is erroneously indicated as Table 1 in the caption.

RESPONSE:

The authors appreciate the reviewer’s comment. Regret the error. Caption improved and Table # was Corrected to Table 3.

All numbers represent the molar ratio of lipids (no units)

2. Paragraph 3.5 At which temperature particle size analysis was performed? How much samples were diluted?

RESPONSE:

The authors appreciate the reviewer’s comment. All determinations were performed in triplicate at room temperature (25°C). The samples were diluted about 100 times. This information has been added to the revised manuscript (section 3.5).

3.Paragraph 3.8 How much volume of PBS medium was employed for the release study?

RESPONSE:

The authors appreciate the reviewer’s comment. PBS (250 ml, pH 7.4) in 250 ml conical flasks was used as a receptor phase. This information has been added to the revised manuscript.

4.How was the cumulative percentage of drug release calculated?

RESPONSE:

The authors appreciate the reviewer’s comment. From a standard curve.

The cumulative amount released at each sampling time is the sum of the amount in the receiver at that time plus the amount in each sample that was removed for analysis (lost due to analysis sampling) and replaced with the buffer.

Results and Discussion:

1.Why results from F4 formulation (control) are not reported in Table 1?

RESPONSE:

The authors appreciate the reviewer’s comment. We agree with the reviewer. We could have reported the results without EE% and DL%. We missed the opportunity. In table 1, we were more concerned about EE= Encapsulation Efficiency; DL= Drug Loading (daunorubicin) and how its encapsulation effected osmolarity. Since F4 was a blank liposome, so we did not include those data on it.

2.Line 150 Table 1 is erroneously referred as Table 2.

RESPONSE:

The authors appreciate the reviewer’s comment. Regret the error, which has been now corrected

  1. The choice of symbols in the graph and denomination of samples in the legend is confusing. I suggest changing the criteria about choosing the symbols for the plot and referring samples as F1, F2 and F3 formulations.

RESPONSE:

The authors appreciate the reviewer’s feedback. In the figure we made changes. The samples are now marked ss F1, F2, F3 and each formulation had a unique color at different temperature 

  1. Paragraph 2.4 Line 242-255 The provided explanation about the reason of higher cytotoxicity for formulation containing cardiolipin is not clear.

RESPONSE:

The authors appreciate the reviewer’s comment. In Paragraph 2.4, we explained our results and findings.

The reason of higher cytotoxicity for formulation containing cardiolipin has been explained in section 2.5

Lines 308-321: “When applying mild hyperthermia, rapid release of DNR was observed from our TSLs formulation. At 4 h, we observed the enhanced effect of CL on DNR accumulations as more DNR accumulated within cells in the case of F1 compared to other formulations. Enrichment of TSLs with CL enhanced the cellular uptake might be due to the ability of CL to form domains on the cell membrane that eventually change its physical stability, fluidity, mechanical structure, allowing more drug to accumulate into cells [29]. Cellular uptake results are consistent with both release and cytotoxicity results.

Cardiolipin (CL) is a negatively charged lipid [22], which can induce lipid structural defects in the cell and mitochondrial membranes [23]. CL can integrate with the cellular lipid domains and induce structural changes in the cell membrane, making the membrane structurally deformed and more permeable [29]. Therefore, in the current study we find that the DNR uptake in the cells by the F1 is significantly higher compared to F2 (no CL) and F3 (formulation similar to DaunoXome).”

5. The authors should demonstrate that cardiolipin form microdomain that can alter the organization of the phospholipid bilayer in liposomes, therefore increasing the cellular uptake and therapeutic outcomes. Alternatively, a sound discussion between the obtained results and the literature available regarding liposomes could be useful. 

RESPONSE:

The authors appreciate the reviewer’s comment. CL enhanced the cellular uptake might be due to the ability of CL to form domains on the cell membrane that eventually change its physical stability, fluidity, mechanical structure, allowing more drug to accumulate into cells [24,29] (section 2.5)

Applying heat will distort liposomes, causing them to release their content rapidly, so we are concerned about CL interaction with cells.

Conclusions:

1.Line 510 It seems from the sentence that the thermosensitive behaviour is related to cardiolipin and not phospholipid compositions. Instead, cardiolipin does not affect the thermosensitive behaviour of liposomes.

RESPONSE:

The authors appreciate the reviewer’s comment. We regret the lack of clarity. We now corrected the statement in the conclusion to read as:

We have designed TSLs using by control of DPPC, MSPC, DSPE-mPEG and cholesterol. The cytotoxicity and cellular uptake of TSLs were further increased by the inclusion of CL.

Reviewer 3 Report

First of all, congratulations on the good work done. It is perfectly structured and very easy to understand. I just wanted to make some comments and if possible solve some doubts. Thanks in advance.

1. In the materials section, it would be necessary to include DSPC (it is not included in the summary either).

 2. In the liposome preparation section, specifically in the hydration phase, why was an ammonium sulfate solution at pH 5.5 selected instead of directly including DNR with the PBS as the hydration medium?

 3. In lines 394 and 395, there is an error, the legend of the table must refer to table3.

 4. Why was the amount of cholesterol decreased in formulation F3?

 5. Most Centigrade (°C) symbols should be corrected.

 6. The titles of sections 3.6 and 3.7 should be in italics, just like the rest of the sections.

 7. In stability studies, is no microbial contamination observed after 1 month of liposome storage? (although protected with N2, from light and at 4°C).

 8. In section 2.2, lines 140 and 150, reference is made to table 2 when discussing the size of the liposomes. I think that those paragraphs should correspond to table 1, which appears above, instead of table 2. In the previous paragraphs, where EE% and DL% data are discussed, it also refers to the same table 1. Although it is true that in table 2 these initial data of size, PI, EE%, DL%, Zeta Potential and Osmolarity are repeated.

 9. Figures 2A and 2B could be included in one, such as figure 4, with the two panels A and B.

 10. The quality of images 5 and 6 (sharpness) should be improved.

 11. In the figure caption of line 334 there is a mistake, it should be Figure 7.

 12. Do you plan to carry out long-term stability studies?

 13. In reference 24, line 583 there is a mistake, the year of publication is duplicated and the doi is collected. In reference 65, line 674, the magazine should go without an abbreviation. In reference 66, line 675, the name of the first author must be capitalized (P.E.)

Author Response

First of all, congratulations on the good work done. It is perfectly structured and very easy to understand. I just wanted to make some comments and if possible solve some doubts. Thanks in advance. The authors appreciate the reviewer’s kind words.

  1. In the materials section, it would be necessary to include DSPC (it is not included in the summary either).

RESPONSE:

The authors appreciate the reviewer’s comment. Distearoylphosphatidylcholine (DSPC) was included in the material section. DSPC was used to prepare liposomes with a similar composition as that of DaunoXome

  1. In the liposome preparation section, specifically in the hydration phase, why was an ammonium sulfate solution at pH 5.5 selected instead of directly including DNR with the PBS as the hydration medium?

RESPONSE:

The authors appreciate the reviewer’s comment. We want to clarify that the transmembrane ammonium sulfate gradients in liposomes produce efficient and stable entrapment of amphipathic weak bases (daunorubicin). This information has been provided in section 2.1, formulation preparation.

  1. In lines 394 and 395, there is an error, the legend of the table must refer to table 3.

RESPONSE:

The authors regret the error. It is now corrected

  1. Why was the amount of cholesterol decreased in formulation F3?

RESPONSE:

The authors appreciate the reviewer’s query. F3 has a similar composition to that of DaunoXome®. We prepared the same lipid ratio using the same lipids compositions.

  1. Most Centigrade (°C) symbols should be corrected. oC

RESPONSE:

The authors regret the error. It is now corrected

  1. The titles of sections 3.6 and 3.7 should be in italics, just like the rest of the sections.

RESPONSE:

The authors regret the error. It is now corrected

  1. In stability studies, is no microbial contamination observed after 1 month of liposome storage? (although protected with N2, from light and at 4°C).

RESPONSE:

The authors appreciate the reviewer’s comment. Our future goals are to conduct in vivo studies including the sterility study.

  1. In section 2.2, lines 140 and 150, reference is made to table 2 when discussing the size of the liposomes. I think that those paragraphs should correspond to table 1, which appears above, instead of table 2. In the previous paragraphs, where EE% and DL% data are discussed, it also refers to the same table 1. Although it is true that in table 2 these initial data of size, PI, EE%, DL%, Zeta Potential and Osmolarity are repeated.

RESPONSE:

The authors regret the error. The Table number now has been corrected to Table 1

  1. Figures 2A and 2B could be included in one, such as figure 4, with the two panels A and B.

RESPONSE:

The authors appreciate this comment. The figures now have been merged

Figures adjusted

  1. The quality of images 5 and 6 (sharpness) should be improved.

RESPONSE:

The authors appreciate this comment. Where there is no drug accumulation, the images appear darker with no cell line impressions. Images were taken using fluorescence microscopy where the intensity and sharpness are correlated to drug accumulation inside cells. For example, our TSLs-CL formulation at 42oC after 6 hours produced a sharp image due to high DNR accumulation.

  1. In the figure caption of line 334 there is a mistake, it should be Figure 7.

RESPONSE:

The authors regret the error. It is now corrected

  1. Do you plan to carry out long-term stability studies?

RESPONSE:

The authors appreciate the reviewer’s comment. Our goal was to develop a stable formulation for in vivo studies. We plan to conduct additional studies.

  1. In reference 24, line 583 there is a mistake, the year of publication is duplicated and the doi is collected. In reference 65, line 674, the magazine should go without an abbreviation. In reference 66, line 675, the name of the first author must be capitalized (P.E.)

RESPONSE:

The authors regret the error. It is now corrected.

Round 2

Reviewer 2 Report

The authors have addressed the reviewer comments and the manuscript is suitable for publication